# Prevalence and determinants of non-communicable diseases risk factors among reproductive aged women of Nepal: Results from Nepal Demographic Health Survey 2016

**Bihungum Bista**[1]*, **Raja Ram Dhungana**[2], **Binaya Chalise**[3], **Achyut Raj Pandey**[4]

**1** Nepal Health Research Council, Kathmandu, Nepal, **2** Institute of Health & Sports, Victoria University, Melbourne, Australia, **3** Graduate School for International Development and Cooperation, Hiroshima University, Higashihiroshima, Japan, **4** DFID/NHSP3/MEOR, Abt Associates, Kathmandu, Nepal

* bistabihungum@gmail.com

## Abstract

### Introduction

Non-Communicable Diseases (NCDs) are the major killer diseases globally. They share the common risk factors such as smoking, harmful use of alcohol, physical inactivity, and low fruits/vegetable consumption. The clustering of these risk factors multiplies the risk of developing NCDs. NCDs affect women inequitably causing significant threats to the health of women and future generations. But, the distribution and clustering of NCDs risk factors among Nepalese women are not adequately explored yet. This study aimed to assess the clustering and socio-demographic distribution of major NCD risk factors in Nepalese women.

### Methods

We used the data of 6,396 women age 15 to 49 years from the recent Nepal Demographic and Health Survey (NDHS). The survey applied a stratified multi-stage cluster sampling method to select the eligible women participants from across Nepal. We analyzed data using the multiple Poisson regression and reported the adjusted prevalence ratio (APR).

### Results

A total of 8.9% of participants were current smokers, 22.2% were overweight and obesity and 11.5% of the participants were hypertensive. Around 6% of participants had co-occurrence of two NCDs risk factors. Smoking, overweight and obesity and hypertension were significantly associated with age, education, province, wealth index, and ethnicity. Risk factors were more likely to cluster in women of age 40-49 years (ARR = 2.95, 95%CI: 2.58–3.38), widow/separated (ARR = 3.09; 95% CI:2.24–4.28) and among Dalit women (ARR = 1.34; 95% CI:1.17–1.55).

**Data Availability Statement:** The data underlying the results presented in the study are available

from (https://dhsprogram.com/data/Using-DataSets-for-Analysis.cfm).

**Funding:** ABT Associates Pty Ltd (Nepal Office) provided support in the form of salary for author Achyut Raj Pandey. There are no patents, products in development or marketed products associated with this research to declare. This does not alter our adherence to PLOS ONE policies on sharing data and materials.

**Competing interests:** ABT Associates Pty Ltd (Nepal Office) provided support in the form of salary for author Achyut Raj Pandey, but did not have any additional role in the study design, data collection and analysis, decision to publish, or preparation of the manuscript. The specific roles of this author is articulated in the 'author contributions' section.

## Conclusion

This study found that NCDs risk factors were disproportionately distributed by age, education, socio-economic status and ethnicity and clustered in more vulnerable groups such as widow/separated women and the Dalit women.

## Introduction

Globally, non-communicable diseases (NCDs) are the number one cause of death and disability. NCDs account for 41 million deaths each year out of which 85% of the deaths occur in low and middle-income countries (LMICs), and nearly half of the deaths (15 million out of 41 million) occur between the age of 30 and 69 years [1–3]. Cardiovascular diseases, cancers, diabetes, and respiratory diseases, also called the 'Group of Four' are responsible for 80% of all NCDs deaths [3]. NCDs are mostly linked with the behavioral (such as tobacco use, harmful use of alcohol, low intake of fruits and vegetables, and physical inactivity) and metabolic (such as obesity, blood sugar, blood pressure, and cholesterol level) risk factors [3] The co-occurrence of two or more of these factors in an individual is referred to as clustering of the risk factors that increase the risk of developing NCDs [4, 5]. Evidence shows that women are more likely to experience the co-occurrence of behavioral and metabolic risk factors thus increasing the risk of NCDs among themselves and in a future generation [6–8]. In Nepal, 15.5% of the population in general, and 11.4% of women reported have three or more risk factors for NCDs [9]. This is rather indication of a higher prevalence of NCDs risk factors in Nepal that may place Nepalese women to the highest disease burden. Compared to men, women also experience fewer symptoms and show less apparent signs of certain NCDs like cardiovascular disease. They are thus less likely to be identified and treated or less likely to be the focus of disease prevention [10]. Furthermore women with NCD risk factors have an adverse impact on their reproductive health as well as in fetal health [11–14]. So, tackling NCDs in women needs a systematic understanding of sociodemographic determinants of to major NCDs risk factors and their clustering [15, 16]. However, in the context of Nepal, there is a paucity of women-focused NCDs studies especially considering social determinants. This study, therefore, aims to assess the magnitude of selected risk factors, individually and in a cluster, and determines their socio-demographic distributions in Nepalese women.

## Methodology

This study used data from the 2016 Nepal Demographic Health Survey (NDHS). NDHS is a periodic survey consisting of a nationally representative sample. The survey used the stratified multi-stage cluster sampling to select individual participants. Initially, 383 primary sampling units (PSU) (wards) were selected based on the probability proportional to PSU size. Then, 30 households per PSU (total 11040 households) were selected using an equal probability systematic selection criterion. A detailed description of the NDHS sampling method is reported elsewhere [17]. The NDHS 2016 adopted a universally standardized DHS questionnaire and measured blood pressure with the validated instrument for the first time in the NDHS series. Blood pressure and anthropometric measurements were only obtained from the systematically selected subsample of 12862 study participants. For this study, we only included 6396 women between 15 and 49 years who had their blood pressure recorded.

## Data collection

**Blood pressure.** Trained enumerator measured blood pressure with UA-767F/FAC (A&D Medical, Tokyo, Japan) blood pressure machines. Enumerators took three readings of blood pressure at the interval of five minutes between each reading and averaged the last two readings to get more accurate blood pressure readings. Participants whose systolic blood pressure (SBP) at the level of 140 mmHg or higher and/or diastolic blood pressure (DBP) of $\geq$90 mmHg or higher or currently taking antihypertensive medicines at the time of data collection were considered hypertensive [17].

**Overweight and obesity.** Weight and height were measured as described in the DHS standard protocol [18]. To calculate body mass index (BMI), weight in kilograms was divided by the height in meter-squared. Women having (BMI $\geq$ 25kg/m$^2$) were categorized as 'overweight and obesity" and the remaining (BMI< 25kg/m$^2$) were categorized as "No overweight and obesity" [17].

**Current tobacco use.** Current tobacco use includes either daily or occasional smoking or use of smokeless tobacco (snuff by mouth, snuff by the nose, chewing tobacco and betel quid with tobacco) [17].

## Explanatory variables

Information related to socio-demographic variables including the age of the participants, ethnicity, educational status, place of residence (rural/urban), province and ecological zone and wealth index were extracted from the NDHS original datasets.

## Statistical analysis

All analyses were performed on STATA 15.1 version using survey (*svy*) set command, defining clusters and sampling weight information. All estimates were weighted by sample weights and presented with 95% confidence intervals (CI). Prevalence estimates were calculated using Taylor series linearization. Chi-square test was used for bivariate analysis to test associations between covariates and dependent variables. Furthermore, multiple Poisson regression was used to calculate the adjusted prevalence ratio (APR) [19, 20]. The numbers of risk factors present within each participant (from 0 to 3) were counted to assess the clustering of risk factors and analyzed using the multiple Poisson regression.

## Ethical consideration

The NDHS 2016 sought ethical approval from the Ethical Review Board (ERB) of the Nepal Health Research Council (NHRC), Nepal and ICF Macro Institutional Review Board, Maryland, USA. Written informed consent was obtained from each participant before enrolling in the survey.

## Results

Table 1 depicts the sociodemographic characteristics of the study participants. Mean age of participants was 29.54±8.92 years and just over half (53.9%) of the participants were 15–29 years. The largest proportion (36.6%) of the participants were from the Janjati group (indigenous group). One third (33.3%) had no formal schooling while 76.6% of the participants were married. Most of the participants belonged to the Terai belt (49.9%) and rural areas (63.3%). Similarly, 22.4% and 20.9% of participants belonged to richer and the richest wealth quintile. Most of the participants were engaged in agriculture or were self-employed.

**Table 1. Socio-demographic distribution of participants.**

| Characteristics | un-weighted count | weighted percent |
|---|---:|---:|
| **Age group** | | |
| 15–29 | 3,498 | 53.9 |
| 30–39 | 1,697 | 27.1 |
| 40–49 | 1,201 | 18.9 |
| **Educational status** | | |
| No education | 2,161 | 33.3 |
| Primary | 1,017 | 16.7 |
| Secondary | 2,324 | 35.5 |
| Higher | 894 | 14.5 |
| **Marital status** | | |
| Never in union | 1,305 | 20.7 |
| Married or living together | 4,919 | 76.6 |
| Widowed/divorced/separated | 172 | 2.7 |
| **Ecological region** | | |
| Mountain | 454 | 6.1 |
| Hill | 2,916 | 44.1 |
| Terai | 3,026 | 49.9 |
| **Residence** | | |
| Rural | 4,129 | 63.0 |
| Urban | 2,267 | 36.9 |
| **Province** | | |
| Province 1 | 909 | 16.8 |
| Province 2 | 1,051 | 19.9 |
| Province 3 | 853 | 22.1 |
| Gandaki | 803 | 9.8 |
| Province 5 | 988 | 16.9 |
| Karnali | 888 | 5.7 |
| Sudurpaschim | 904 | 8.8 |
| **Wealth index** | | |
| Poorest | 1,347 | 16.9 |
| Poorer | 1,304 | 19.1 |
| Middle | 1,319 | 20.6 |
| Richer | 1,319 | 22.4 |
| Richest | 1,107 | 20.9 |
| **Occupational status** | | |
| Did not work | 2,003 | 32.3 |
| Services | 863 | 15.0 |
| Agriculture/ self-employed | 3,196 | 46.9 |
| Manual | 331 | 5.8 |
| **Ethnic group** | | |
| Advantage group | 2,254 | 31.3 |
| Dalit | 851 | 12.6 |
| Janjati | 2,268 | 36.6 |
| Other | 1,023 | 19.5 |
| **Total** | 6,396 | 100 |

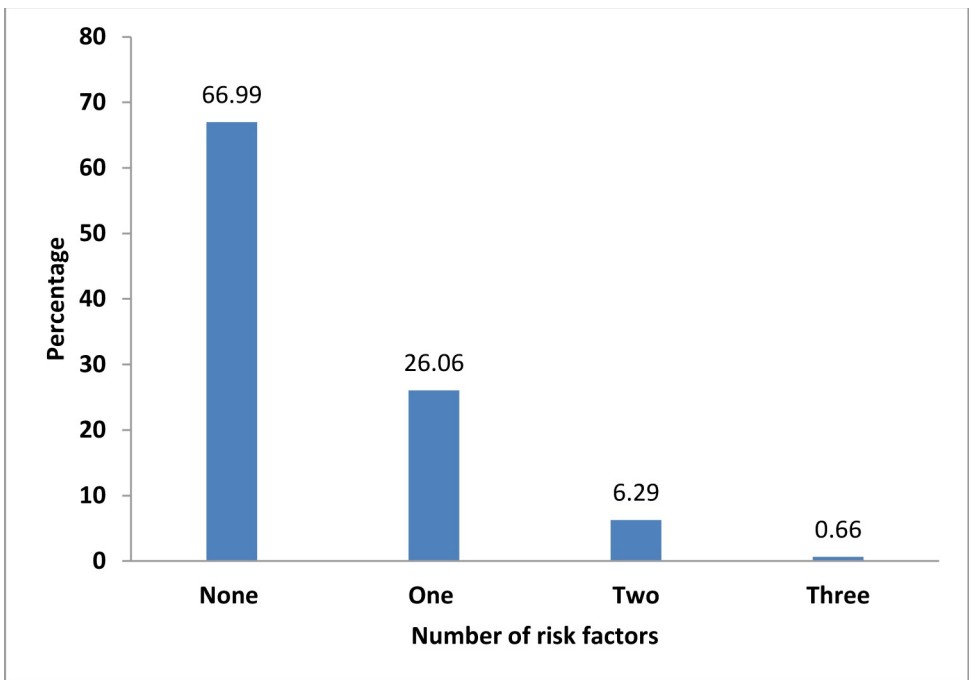

**Fig 1. Prevalence of number of NCDs risk factors among participants.**

Fig 1 shows the NCDs risk factor prevalence by the number of factors. More than one-fourth of the participants had one NCDs risk factor and 6.3% of participants had two NCD risk factors.

## Distribution of non-communicable diseases risk factors

The prevalence of current tobacco use was 8.9%. Women of age 40–49 years (22.4%), with no education (18.8%) and widowed/divorced/separated women (29.1%) had the highest prevalence of current tobacco use as indicated in "Table 2". Similarly, current tobacco use was significantly associated with the ecological zone, province, wealth index, occupation, and ethnicity "Table 2".

The adjusted multivariate model shows significantly higher prevalence of tobacco use among women of 40–49 years of age (APR: 3.70; 95% CI: 2.65–5.17), having no education, widowed/divorced/separated (APR: 1.04; 95% CI:1.4–3.98), from province one, in the lowest wealth quintile and among women from Dalit ethnic/caste group (APR:1.68; 95% CI:1.27–2.23). However, the prevalence of tobacco use was significantly lower among poor women (APR: 0.69; 95% CI:0.55–0.86) residing on province 5 (APR: 0.64; 95% CI: 0.45–0.90)) "Table 3".

The prevalence of overweight and obesity/obesity was 22.2%., which was significantly high in women age 40–49 years compared to that of 15–29 years (11.9%) women "Table 2". The prevalence of overweight and obesity significantly varied by education status "Table 2". Compared to never union, the prevalence of overweight and obesity was significantly higher among married/ living together women (27.3%) and divorced/widowed/separated women (25.5%). In multivariable analysis, the prevalence of Overweight and obesity was significantly higher in the elder age group (APR:1.97; 95% CI:1.68–2.31), married women (APR:4.02; 95% CI: 2.98–5.40), and those women belonging to wealthiest quintile (APR-3.38; 95% CI-2.63–4.34). However,

**Table 2. Prevalence (%) of non-communicable diseases risk factors among 15–49 years women.**

| Characteristics | Current tobacco use | | Overweight and obesity | | Hypertension | |
|---|---|---|---|---|---|---|
| | n | Prevalence | n | Prevalence | n | Prevalence |
| **Age(yrs)** | | | | | | |
| 15–29 | 3,498 | 2.6 [1.9–3.4] | 3,169 | 11.9 [10.6–13.5] | 3,498 | 4.0 [3.3–4.9] |
| 30–39 | 1,697 | 12.1 [10.3–14.1] | 1,647 | 33.3 [29.9–36.8] | 1,697 | 13.1 [11.3–15.1] |
| 40–49 | 1,201 | 22.4 [19.3–25.8] | 1,197 | 34.2 [30.3–38.4] | 1,201 | 24.9 [21.9–28.3] |
| *P-value* | | <0.001 | | <0.001 | | <0.001 |
| **Educational level** | | | | | | |
| no education | 2,161 | 18.8 [16.6–21.2] | 2,073 | 19.9 [17.6–22.4] | 2161 | 12.9 [11.1–14.8] |
| Primary | 1,017 | 9.9 [7.8–12.4] | 936 | 27.8 [24.4–31.6] | 1,017 | 12.39 [10.4–14.7] |
| secondary | 2,324 | 2.5 [1.8–3.3] | 2,173 | 20.4 [17.7–23.3] | 2,324 | 7.8 [6.6–9.3] |
| Higher | 894 | 0.7 [0.3–1.4] | 831 | 27.0 [22.9–31.7] | 894 | 9.1 [6.4–12.7] |
| *P-value* | | <0.001 | | <0.001 | | <0.001 |
| **Marital status** | | | | | | |
| never in union | 1,305 | 1.8 [0.9–3.4] | 1,305 | 5.3 [3.9–6.9] | 1305 | 2.9 [2.1–4.1] |
| married or living together | 4,919 | 10.1 [9.0–11.3] | 4,537 | 27.3 [25.0–29.7] | 4919 | 12.2 [10.9–13.6] |
| widowed/divorced/separated | 172 | 29.1 [21.1–38.6] | 171 | 25.5 [18.2–34.3] | 172 | 16.9 [11.4–24.2] |
| *P-value* | | <0.001 | | <0.001 | | <0.001 |
| **Ecological zone** | | | | | | |
| Mountain | 454 | 14.7 [10.4–19.9] | 412 | 20.7 [15.0–27.7] | 454 | 10.6 [7.2–15.5] |
| Hill | 2,916 | 10.9 [9.2–13.1] | 2,776 | 26.9 [23.9–30.2] | 2916 | 12.2 [10.4–14.3] |
| Terai | 3,026 | 6.4 [5.3–7.6] | 2,825 | 18.5 [16.4–20.7] | 3026 | 8.8 [7.7–10.1] |
| *P-value* | | <0.001 | | <0.001 | | 0.008 |
| **Residence** | | | | | | |
| Urban | 4,129 | 8.5 [7.1–10.2] | 3,892 | 26.3 [23.7–29.0] | 4,129 | 11.0 [9.6–12.6] |
| Rural | 2,267 | 9.6 [8.18–11.21] | 2,121 | 15.6 [13.7–17.9] | 2,267 | 9.5 [8.1–11.2] |
| *P-value* | | 0.334 | | <0.001 | | 0.171 |
| **Province** | | | | | | |
| Province 1 | 909 | 10.8 [8.6–13.5] | 863 | 27.6 [23.6–32.0] | 909 | 10.7 [8.8–13.1] |
| Province 2 | 1,051 | 3.0 [2.0–4.5] | 953 | 10.9 [8.9–13.5] | 1,051 | 6.6 [5.4–8.1] |
| Province 3 | 853 | 10.1 [7.2–14.0] | 815 | 34.9 [29.5–40.6] | 853 | 13.3 [10.2–17.2] |
| Gandaki | 803 | 10.1 [7.4–13.8] | 774 | 31.7 [27.7–35.9] | 803 | 15.4 [12.5–18.8] |
| Province 5 | 988 | 7.5 [5.6–9.9] | 930 | 18.8 [15.7–22.3] | 988 | 11.9 [9.5–14.8] |
| Karnali | 888 | 15.9 [13.1–19.3] | 833 | 10.6 [7.8–14.2] | 888 | 7.4 [5.4–10.2] |
| Sudurpaschim | 904 | 12.4 [10.0–15.3] | 845 | 9.1 [5.8–14.2] | 904 | 5.1 [3.55–7.14] |
| *P-value* | | <0.001 | | <0.001 | | <0.001 |
| **Wealth index** | | | | | | |
| Poorest | 1,347 | 19.5 [17–22.3] | 1,265 | 10.0 [8.1–12.4] | 1347 | 8.3 [6.6–10.5] |
| Poorer | 1,304 | 10.7 [9.1–12.6] | 1,215 | 15.6 [13.5–18.0] | 1304 | 10.8 [8.9–12.9] |
| Middle | 1,319 | 6.2 [4.9–7.8] | 1,227 | 14.1 [11.8–16.7] | 1319 | 9.0 [7.5–10.9] |
| Richer | 1,319 | 6.6 [4.1–10.4] | 1,246 | 23.4 [20.8–26.3] | 1,319 | 8.49 [6.8–10.5] |
| Richest | 1,107 | 3.71 [2.4–5.7] | 1,060 | 44.9 [41.1–48.8] | 1107 | 15.4 [13.2–17.8] |
| *P-value* | | <0.001 | | <0.001 | | <0.001 |
| **Occupation*** | | | | | | |
| Did not work | 2,003 | 4.4 [3.5–5.6] | 1,826 | 24.1 [21.7–26.8] | 2,003 | 9.92 [8.6–11.5] |
| Services | 863 | 6.4 [4.3–9.4] | 836 | 39.5 [34.8–44.4] | 863 | 14.2 [11.1–17.9] |
| Agriculture(self-employed) | 3,196 | 12.4 [10.94–13.98] | 3,035 | 14.6 [13.0–16.3] | 3,196 | 9.5 [8.3–10.9] |
| Manual | 331 | 12.6 [8.7–17.8] | 313 | 30.9 [23.8–38.9] | 331 | 11.2 [7.7–16.1] |

*(Continued)*

**Table 2.** (Continued)

| Characteristics | Current tobacco use | | Overweight and obesity | | Hypertension | |
|---|---|---|---|---|---|---|
| | n | Prevalence | n | Prevalence | n | Prevalence |
| *P-value* | | <0.001 | | <0.001 | | 0.014 |
| **Ethnicity** | | | | | | |
| Advantage group | 2,254 | 7.2 [6.1–8.6] | 2,142 | 24.5 [21.1–28.3] | 2254 | 9.8 [8.33–11.62] |
| Dalit | 851 | 14.9 [12.1–18.3] | 782 | 18.5 [15.3–22.0] | 851 | 10.7 [8.5–13.3] |
| Janjati | 2,268 | 11.3 [9.6–13.3] | 2,146 | 26.6 [23.3–30.1] | 2268 | 12.1 [10.5–14.00] |
| Others | 1,023 | 3.2 [2.21–4.6] | 943 | 13.2 [10.9–15.8] | 1,023 | 8.1 [6.64–9.72] |
| *P-value* | | <0.001 | | <0.001 | | 0.005 |
| **Total** | 6396 | 8.9 [7.9–10.1] | 6,013 | 22.2 [20.5–24.0] | 6396 | 10.4 [9.4–11.7] |

*10 cases missing

women residing in Sudurpaschim province (APR: 0.42; 95% CI: 0.38–0.71) and employed on agriculture had lower (APR: 0.71; 95% CI: 0.62–0.82) had lower prevalence of overweight and obesity "Table 3".

The prevalence of hypertension was 10.5%. It significantly varied by the age of participants, For instance, 40–49 years participants had the highest rate of hypertension. Secondary education was significantly associated with a higher prevalence of hypertension compared to primary and no education. Likewise, the rate of hypertension was also significantly different in the province, wealth index, occupation, and ethnicity "Table 2".

Multivariable analysis shows that higher prevalence of hypertension on elder age group women (APR; 5.73;95% CI: 4.25–7.7) among married women (APR: 1.97; 95% CI: 1.35–1.68), belonging to the wealthiest group (APR:1.45; 95% CI: 1.00–2.09), among Dalit women (APR;1.47; 95% CI: 1.09–1.97), and among the Janjati women (APR: 1.28; 95% CI: 1.04–1.57). But hypertension was less prevalent (APR: 0.58; 95% CI: 0.37–0.89) among women residing on Sudurpaschim province and engaged in agriculture (APR: 0.78; 95% CI: 0.62–0.82) "Table 3".

## Multivariable analysis of socio-demographic characteristics with noncommunicable diseases risk factors

Women of 40–49 years were more likely to experience NCD risk factors than 15–29 years aged women (ARR: 2.95; 95% CI: 2.58–3.38) "Table 4". Compared to the women with no education, women who had pursued a secondary level of education were less likely (ARR: 0.87; 95% CI: 0.77–0.98) to experience NCD risk factors. The adjusted risk ratio for married and widowed/divorced/separated women was almost 3 times (ARR: 2.91; 95% CI: 2.77–3.74) and (ARR: 3.09; 95% CI: 2.24–4.28) than that of women who had never in a union. Similarly, the richest women were more likely (ARR: 1.5; 95% CI: 1.27–1.77) to suffer from NCDs risk factors in comparison to the poorest women. Furthermore, women employed in the agriculture sector were less likely (ARR: 0.83; 95% CI: 0.75–0.92) to suffer from NCD risk factors than women who were not employed. Regarding ethnicity, Dalit women were more likely (ARR: 1.34; 95% CI: 1.17–1.55) to have NCD risk factors in comparison to advantage group "Table 4".

## Discussion

NCDs have different consequences for women in comparison to men [21]. In resource-challenged settings like Nepal where diagnosis and care for NCDs are less accessible and affordable to women prominently due to patriarchal society beliefs as well as limited health

**Table 3. Relationship of socio-demographic characteristics with non-communicable disease risk factors.**

| | Current tobacco use APR | Overweight and obesity APR | Hypertension APR |
|---|---|---|---|
| **Age group (Years)** | | | |
| 15–29 | 1 | 1 | 1 |
| 30–39 | 2.46 [1.77–3.43]*** | 1.85 [1.60–2.13]*** | 2.8 [2.09–3.76]*** |
| 40–49 | 3.7 [2.65–5.17***] | 1.97 [1.68–2.31]*** | 5.73 [4.25–7.71]*** |
| **Educational status** | | | |
| No education | 1 | 1 | 1 |
| Primary | 0.71 [0.57–0.88]** | 1.27 [1.10–1.46]** | 1.28 [1.03–1.59]* |
| Secondary | 0.28 [0.20–0.40]*** | 1.09 [0.94–1.25] | 1.2 [0.88–1.62] |
| Higher secondary level or more | 0.09 [0.04–0.22]*** | 1.12 [0.93–1.36] | 1.31 [0.90–1.91] |
| **Marital status** | | | |
| Never in union | 1 | 1 | 1 |
| Married or living together | 1.37 [0.75–2.49] | 4.02 [2.98–5.40]*** | 1.97 [1.35–2.89]*** |
| Widowed/divorced/separated | 2.03 [1.04–3.98]* | 3.29 [2.06–5.25]*** | 1.91 [1.11–3.30]* |
| **Ecological region** | | | |
| Mountain | 1 | 1 | 1 |
| Hill | 1.01 [0.72–1.43] | 0.8 [0.57–1.11] | 0.79 [0.55–1.13] |
| Terai | 1.19 [0.79–1.79] | 0.71 [0.50–1.01] | 0.71 [0.48–1.06] |
| **Residence** | | | |
| Rural | 1 | 1 | 1 |
| Urban | 1.16 [0.96–1.41] | 0.98 [0.85–1.13] | 0.94 [0.75–1.16] |
| **Province** | | | |
| Province 1 | 1 | 1 | 1 |
| Province 2 | 0.28 [0.17–0.46]*** | 0.46 [0.36–0.58]*** | 0.61 [0.43–0.87]** |
| Province 3 | 1 [0.72–1.39] | 0.9 [0.76–1.07] | 1.1 [0.80–1.51] |
| Gandaki | 0.92 [0.67–1.26] | 1 [0.84–1.19] | 1.3 [0.93–1.82] |
| Province 5 | 0.64 [0.45–0.90]** | 0.71 [0.59–0.86]*** | 1.2 [0.89–1.63] |
| Karnali | 1.02 [0.75–1.39] | 0.52 [0.38–0.71]*** | 0.81 [0.53–1.25] |
| Sudurpaschim | 0.89 [0.66–1.21] | 0.42 [0.28–0.63]*** | 0.58 [0.37–0.89]* |
| **Wealth index** | | | |
| Poorest | 1 | 1 | 1 |
| Poorer | 0.69 [0.55–0.86]*** | 1.58 [1.27–1.97]*** | 1.34 [1.00–1.79] |
| Middle | 0.51 [0.38–0.68]*** | 1.61 [1.23–2.12]*** | 1.22 [0.88–1.69] |
| Richer | 0.52 [0.34–0.81]** | 2.32 [1.80–2.97]*** | 1.04 [0.72–1.48] |
| Richest | 0.37 [0.22–0.60]*** | 3.38 [2.63–4.34]*** | 1.45 [1.00–2.09]* |
| **Occupational status** | | | |
| Did not work | 1 | 1 | 1 |
| Services | 1.5 [0.98–2.27] | 1.05 [0.93–1.19] | 1.02 [0.81–1.28] |
| Agriculture(self-employed) | 1.3[0.97–1.74] | 0.71[0.62–0.82]*** | 0.78[0.64–0.96]* |
| Manual | 1.4 [0.93–2.11] | 0.9 [0.73–1.12] | 0.78 [0.53–1.16] |
| **Ethnic group** | | | |
| Advantage group | 1 | 1 | 1 |
| Dalit | 1.68 [1.27–2.23]*** | 1.09 [0.86–1.36] | 1.47 [1.09–1.97*] |
| Janjati | 1.24 [0.98–1.57] | 1.1 [0.97–1.26] | 1.28 [1.04–1.57*] |
| Others | 0.78 [0.49–1.26] | 0.82 [0.67–1.02] | 1.34 [0.97–1.86] |

*** significant at p-value < 0.001.

**significant at p-value < 0.01.

* significant at p-value < 0.05.

**Table 4. Mean number of NCD risk factors and multivariable analysis of clustering of NCD risk factors.**

| Characteristics | Mean number | Clustering of NCD risk factors Adjusted Risk Ratio(ARR) |
|---|---|---|
| **Age group(in yrs)** | | |
| 15–29 | 1.1[1.1,1.1] | |
| 30–39 | 1.2[1.2,1.3] | 2.16[1.90–2.46]*** |
| 40–49 | 1.3[1.3,1.4] | 2.95[2.58–3.38]*** |
| **Educational status** | | |
| No education | 1.3[1.2,1.3] | |
| Primary | 1.3[1.2,1.4] | 1.07[0.97–1.19] |
| Secondary | 1.2[1.2,1.2] | 0.87[0.77–0.98]* |
| Higher | 1.2[1.1,1.3] | 0.92[0.78–1.08] |
| **Maritial status** | | |
| Never in union | 1.1[1.0,1.1] | |
| Married or living together | 1.2[1.2,1.3] | 2.91[2.27–3.74]*** |
| Widowed/divorced/separated | 1.3[1.2,1.4] | 3.09[2.24–4.28]*** |
| **Ecological region** | | |
| Mountain | 1.3[1.2,1.3] | |
| Hill | 1.3[1.2,1.3] | 0.88[0.74–1.04] |
| Terai | 1.2[1.2,1.3] | 0.85[0.70–1.03] |
| **Residence** | | |
| Rural | 1.3[1.2,1.3] | |
| Urban | 1.2[1.2,1.3] | 1.01[0.91–1.12] |
| **Province** | | |
| province 1 | 1.2[1.2,1.3] | |
| province 2 | 1.2[1.1,1.2] | 0.45[0.37–0.55]*** |
| province 3 | 1.3[1.2,1.3] | 0.99[0.87–1.12] |
| Gandaki | 1.3[1.2,1.4] | 1.07[0.92–1.23] |
| province 5 | 1.2[1.2,1.2] | 0.8[0.69–0.93]** |
| Karnali | 1.2[1.1,1.2] | 0.73[0.62–0.86]*** |
| Sudurpaschim | 1.1[1.1,1.1] | 0.61[0.51–0.74]*** |
| **Wealth index** | | |
| Poorest | 1.2[1.1,1.2] | |
| Poorer | 1.3[1.2,1.3] | 1.05[0.92–1.18] |
| Middle | 1.2[1.2,1.3] | 0.93[0.78–1.10] |
| Richer | 1.2[1.2,1.3] | 1.1[0.94–1.30] |
| Richest | 1.3[1.2,1.3] | 1.5[1.27–1.77]*** |
| **Occupational status** | | |
| Did not work | 1.3[1.2,1.3] | |
| Services | 1.3[1.2,1.3] | 1.09[0.97–1.22] |
| Agriculture(self-employed) | 1.2[1.2,1.2] | 0.83[0.75–0.92]*** |
| Manual | 1.2[1.2,1.3] | 0.94[0.78–1.12] |
| **Ethnic group** | | |
| Advantage group | 1.2[1.1,1.2] | |
| Dalit | 1.3[1.3,1.4] | 1.34[1.17–1.55]*** |
| Janjati | 1.3[1.2,1.3] | 1.16[1.05–1.28]** |
| Other | 1.2[1.1,1.2] | 0.95[0.80–1.13] |

*** significant at p-value < 0.001.

**significant at p-value < 0.01.

* significant at p-value < 0.05.

infrastructure, and human-resource capacity [22]. As a result, NCDs are often detected at the later stage of a woman's life with a consequence of premature death. So, this study attempted to identify NCDs risk factors associated with women. This information could be useful in designing preventative strategies against NCDs risk factors.

Our study demonstrated that the proportion of tobacco use was nearly threefold higher in 30–40 years age group women which have also been observed in previous studies [23, 24]. Higher prevalence of tobacco use in older age group may be understood on light of low level of awareness/education and means of stress coping strategies in comparison to elder age group women. High prevalence of tobacco use in women with childbearing age deserves attention because of its adverse maternal and child health outcomes in the perinatal period [25]. Higher prevalence of tobacco use among divorced women than currently married, which is in line with previous studies [23, 26], it might be because of stress coping strategy or an option to overcome loneliness.

The study revealed the poorest wealth quintile as a key determinant of tobacco use while the prevalence of hypertension was more among participants of the highest wealth quintile. An increase in taxation could be one of the potential strategies to control tobacco use. Evidence suggests that around 10% increase in tobacco price reduces smoking by about 8% in low- and middle-income countries and by 4% in high-income countries [27]. Such strategy could be especially effective in the poorest segment of the population.

We observed the increasing trend of hypertension and overweight and obesity with increasing age and economic status. This seems to be a usual phenomenon as reported in other studies from different settings [4, 9]. On the other hand, it could be due to the reduced level of physical activity as people grew older and wealth status. The prevalence of Overweight and obesity in reproductive-age women has nearly tripled from 9% in the last ten years in Nepal [17, 28, 29]. This finding alarms the focus of maternal and child health programmes on NCDs risk factors like maternal obesity, due to their adverse consequences on maternal and child health. Maternal obesity can substantially interfere the fetal development and determines the long term health of the offspring [30]. It is also a major risk factor for gestational diabetes, pre-eclampsia and pregnancy-induced hypertension in women [31, 32]. Our study demonstrated a higher prevalence of hypertension in Gandaki province and the lower in Sudurpaschim province that is in line with national findings carried out in the general population [33]. Differences in the level of physical activity associated with occupational practices, dietary patterns, might have attributed the higher prevalence of hypertension in Gandaki province compared to other provinces. Furthermore higher level of urbanization and sedentary lifestyle in Gandaki province and Province 3 in comparison to other provinces may have accounted higher prevalence of hypertension.

Clustering of NCDs risk factors seems to be more with growing age, among well-off, and in Dalits and Janajatis -known as the disadvantaged ethnic groups in Nepal. Previous studies from multiple other countries have also found that the clustering of risk factors becomes increasingly common with increasing age [4, 5, 34]. As Nepal has been witnessing a rapid increase in life expectancy and the median age of the population, the problems can escalate in the coming years [35]. The country may need additional investment in prevention as well as long term care for NCDs to cater to the need of the geriatric population. Moreover, NCDs are considered to have a serious impact on the economic growth of the country. Reducing NCDs by 5–10% is thus a development agenda rather than a health problem confining it under the jurisdiction of the health sector [36]. This calls for multisectoral actions with coordinated efforts of the health sector.

Similarly, this study depicts the odds of clustering of NCDs risk factors higher among the wealthiest women which were also observed in the previous study in Bhutan [37]. Similar to

individual risk factors like obesity and hypertension, the clustering of NCDs risk factors in the wealthier group can be linked with the adoption of a sedentary lifestyle. Similar factors might also be responsible for higher odds for the clustering of risk factors in province 1, province 3 and Gandaki province. Additionally, the pace of urbanization and westernization of dietary patterns might also have a role in the clustering of risk factors in specific provinces. Women who have a secondary level of education had a lower risk of clustering of NCDs risk factors which contradicts the findings from Bangladesh [38]. The difference in evidence may be due to differences in NCDs prevention and control contents in secondary level education. Furthermore, women involved in agriculture (self-employed), which generally involve vigorous physical activity, sector have low odds of clustering of NCDs risk factors. Vigorous physical activity is a protective factor against obesity and it is expected to reduce the risk of clustering NCDs risk factors [39].

Being cross-sectional in nature, the study does not establish causality. The NDHS 2016 mainly focused on maternal and child health issues, thus the NDHS did not measured other important biomarkers of NCDs risk such as elevated blood pressure, blood sugar and cholesterol level. The lack of information on biomarkers limited this to reveal the evidence around NCDs risk factors with sufficient depth.

## Conclusion

Similar to most other NCD risk factors, clustering NCD risk factors seem to be more common in the richer segment of the population and higher age group among women. Nepal, that has been facing epidemiological transition with the increasing burden of NCDs while communicable diseases, maternal and neonatal conditions still bear the significant burden, need to make careful choices of the cost-effective interventions.

## Author Contributions

**Conceptualization:** Bihungum Bista, Achyut Raj Pandey.

**Formal analysis:** Bihungum Bista.

**Supervision:** Achyut Raj Pandey.

**Validation:** Achyut Raj Pandey.

**Writing – original draft:** Bihungum Bista, Raja Ram Dhungana, Binaya Chalise.

**Writing – review & editing:** Bihungum Bista, Raja Ram Dhungana, Binaya Chalise, Achyut Raj Pandey.

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
