## [Decision Letter · Decision Letter 0]

19 Aug 2019

PONE-D-19-15867

Socio-demographic correlates and clustering of Non-Communicable Diseases risk factors among reproductive aged women of Nepal: Results from Nepal Demographic Health Survey 2016

PLOS ONE

Dear Mr Bista,

Thank you for submitting your manuscript to PLOS ONE. After careful consideration, we feel that it has merit but does not fully meet PLOS ONE’s publication criteria as it currently stands. Therefore, we invite you to submit a revised version of the manuscript that addresses the points raised during the review process.

The manuscript has been sent to two reviewers and their comments are appended below. The reviewers have raised number of concerns including the methodology and discussion of results. Hope the comments would be very useful to revise your manuscript. 

We would appreciate receiving your revised manuscript by Oct 03 2019 11:59PM. To enhance the reproducibility of your results, we recommend that if applicable you deposit your laboratory protocols in protocols.io, where a protocol can be assigned its own identifier (DOI) such that it can be cited independently in the future. For instructions see: http://journals.plos.org/plosone/s/submission-guidelines#loc-laboratory-protocols

We look forward to receiving your revised manuscript.

Kind regards,

Kannan Navaneetham

Academic Editor

PLOS ONE

Journal Requirements:

2. We note that you have stated that “The funders had no role in study design, data collection and analysis, decision to publish, or preparation of the manuscript” in your financial disclosure. Please also provide the names of the funders in your financial disclosure.

3. Please carefully proofread your manuscript to correct any typographical errors. For example on line 39 “Dalit and Janajati. .” Should be “Dalit and Janajati.” And on line 44 “non-communicable disease (NCDs)” should be “non-communicable diseases (NCDs)”.

4. Please update your data availability statement to explain how researchers can access the data used in this study. The current statement is incomplete “contact via XXX”.

5. In your ethics statement please clarify whether you obtained ethics approval to conduct this study. If you did not obtain ethics approval for this secondary analysis of NDHS data please state this. Please also clarify whether the authors had access to any identifying information.

6. Thank you for stating the following in the Financial Disclosure section:

he funders had no role in study design, data collection and analysis, decision to publish, or preparation of the manuscript.

We note that one or more of the authors are employed by a commercial company: Abt Associates

7. Please include a caption for figure 1.

8. We note you have included a table to which you do not refer in the text of your manuscript. Please ensure that you refer to Table 1 in your text; if accepted, production will need this reference to link the reader to the Table.

Reviewers' comments:

Reviewer's Responses to Questions

**Comments to the Author**

1. Is the manuscript technically sound, and do the data support the conclusions?

Reviewer #1: Partly

Reviewer #2: Partly

2. Has the statistical analysis been performed appropriately and rigorously? 

Reviewer #1: Yes

Reviewer #2: No

3. Have the authors made all data underlying the findings in their manuscript fully available?

Reviewer #1: No

Reviewer #2: Yes

4. Is the manuscript presented in an intelligible fashion and written in standard English?

Reviewer #1: No

Reviewer #2: Yes

5. Review Comments to the Author

Reviewer #1: Comments to the author

Title: Socio demographic correlates and clustering of non-communicable disease risk factors among reproductive aged women in Nepal: Results from the Nepal Demographic Health Survey.

General comments: The research article addresses a very important topic within the context of a developing country experiencing epidemiological transition. Clustering of NCD risk factors predisposes women to the risk of developing NCDs. The article is poorly written but with much improvement it has the potential to be published. There is need to consider language usage, thus the authors should not use colloquial language but statements should be supported by references to enhance authenticity of claims. The discussion section should be concise and provide reasons for any differences in findings when compared to other studies elsewhere within similar context. The authors have not followed the journal’s referencing style.

Abstract

The structure of the abstract does not follow the journal requirements. The abstract needs to be structured into the following;

• Background

• Methods

• Results

• Conclusion

Page 2 Line 31: The authors have inappropriately used the word similarly.

Page 2 line 36: Dalit) - correction has to be made.

Page 2 line 39: In the conclusion section the authors indicate that the clustering of NCD risk factors was more vulnerable groups such as widow/separated, Dalit and Janajati although for Janajati they have not shown prevalence ratios. Again there are two full stops (..).Correct that.

Introduction

Check on the referencing style…the reference number come before a full stop (period), not after.

Page 3 Line 50, 51: There is textual overlap with previously published work on the sentence; …. NCDs share the common risk factors such as low intake of fruit and vegetables, low level of physical activity, tobacco use, harmful use of alcohol, obesity, raised blood pressure, raised blood cholesterol and glucose. As a result there is need for citation and paraphrasing.

Page 3 Lines 53-60 there is need for language editing. For instance the sentence,… In context of Nepal, STEPS survey 2013, reported that 15.5% of general population and 11.4% of women had three or more risk factors of NCD in them Can be corrected to read as In Nepal, the STEPS survey 2013 indicated that 15.5% in the general population and 11.4% of women reported three or more risk factors for NCDs.

Page 3 Line 61-62: There is need for citation for this statement.

The general observation is that there is need to consider language usage, thus the authors should not use colloquial language but statements should be supported by references to enhance authenticity of claims. This section can be improved.

Methodology

The methodology for the study is well explained. Perhaps a question one would ask is that since the NDHS used a multi stage stratified sampling, how were cluster and sample design effects dealt with in the analysis of data.

Results

Page 12, Line 110: Make a correction to the first sentence ‘Just over half (53.95%) of the participants were of aged 15-29 years, the highlighted word should be ages. Make similar corrections across the result section and ensure that proper language is used for interpretation of results.

Page 14, Line 121: 26.08%, you can’t stand the sentence with a number. Kindly make correction. Moreover its ‘one risk factor’ Not ‘One risk factors’. There is need for language editing to remove typing and grammatical errors.

There is an over-use and at times misuse of the word ‘similarly’.

For interpretation of table 3, insert confidence intervals for the adjusted prevalence ratios (APR) e.g page 24 line 155, put in confidence intervals together with APRs for primary, secondary or higher education).

Under the subsection ‘Overweight’ page 24, Line 164-170 there are many typos, please make language corrections.

Discussion

Page 26 line 201-201 the sentence is not clear….So, this study aim to identify at risk women to possess NCD risk factors…. You need to rephrase the sentence.

There is no need to categorise the discussion into sub headings. The authors need only to discuss key findings, what are unique and important findings of the study. Only discuss such…what is new and emerging? There are six pages on the discussion section; it shows lack of focus and discussion of salient issues. The discussion section should be concise and provide reasons for any differences in findings when compared to other countries. There should be a subsection on strengths and limitations of the study.

References

The authors have not followed the journal’s referencing style. They use the APA style in the reference section, while in the text they use AMA.

Reviewer #2: Investigating the clustering of non-communicable diseases risk factors is important in the wake of NCDs epidemic in low and middle income countries. The strength of this study is its national representativeness. However, I doubt if there is sufficient data to explore clustering in this study population. Whereas, there are 8 common NCD risk factors – Smoking, Alcohol intake, Unhealthy diet and physical inactivity (behavioural factors), obesity, hypertension, hyperglycaemia and hyperlipidaemia (biological factors). Authors explored just three of these factors they also did not clearly define what clustering is in their study. Although, the research has merit, the paper is poorly written as it is.

1.Is the manuscript technically sound, and do the data support the conclusions?

Authors need to do more work on the manuscript. - They shoulsd make reference to articles published on the same topic particularly the clustering of NCD risk factors

2. Has the statistical analysis been performed appropriately and rigorously?

Further comments in the attachment

3.Have the authors made all data underlying the findings in their manuscript fully available?

Not particularly but the data is in the public domain

4. Is the manuscript presented in an intelligible fashion and written in standard English?

Needs improvement

6. PLOS authors have the option to publish the peer review history of their article (what does this mean?). If published, this will include your full peer review and any attached files.

Reviewer #1: Yes: Mpho Keetile,PhD

Reviewer #2: No

---

## [Author Response · Author response to Decision Letter 0]

9 Oct 2019

RESPONSE TO REVIEWER 1

Comments to the author

Title: Socio demographic correlates and clustering of non-communicable disease risk factors among reproductive aged women in Nepal: Results from the Nepal Demographic Health Survey.

General comments: The research article addresses a very important topic within the context of a developing country experiencing epidemiological transition. Clustering of NCD risk factors predisposes women to the risk of developing NCDs. The article is poorly written but with much improvement it has the potential to be published. There is need to consider language usage, thus the authors should not use colloquial language but statements should be supported by references to enhance authenticity of claims. The discussion section should be concise and provide reasons for any differences in findings when compared to other studies elsewhere within similar context. The authors have not followed the journal’s referencing style.

Abstract

The structure of the abstract does not follow the journal requirements. The abstract needs to be structured into the following;

• Background

• Methods

• Results

• Conclusion

Response- Correction has been made in revised manuscript.

Page 2 Line 31: The authors have inappropriately used the word similarly.

Response- Correction has been made in revised manuscript.

Page 2 line 36: Dalit) - correction has to be made.

Response- Correction has been made in revised manuscript.

Page 2 line 39: In the conclusion section the authors indicate that the clustering of NCD risk factors was more vulnerable groups such as widow/separated, Dalit and Janajati although for Janajati they have not shown prevalence ratios. Again there are two full stops (..).Correct that.

Response: Thank you. Correction has been made.

Introduction

Check on the referencing style…the reference number come before a full stop (period), not after.

Response: Thank you for your comment. It has been made amended on revised manuscript.

Page 3 Line 50, 51: There is textual overlap with previously published work on the sentence; …. NCDs share the common risk factors such as low intake of fruit and vegetables, low level of physical activity, tobacco use, harmful use of alcohol, obesity, raised blood pressure, raised blood cholesterol and glucose. As a result there is need for citation and paraphrasing.

Response: Thank you.Sentence has been paraphrased and reference has been added on revised manuscript.

Page 3 Lines 53-60 there is need for language editing. For instance the sentence,… In context of Nepal, STEPS survey 2013, reported that 15.5% of general population and 11.4% of women had three or more risk factors of NCD in them Can be corrected to read as In Nepal, the STEPS survey 2013 indicated that 15.5% in the general population and 11.4% of women reported three or more risk factors for NCDs. 

Response: Thank you. Sentence has recomposed as advised.

Page 3 Line 61-62: There is need for citation for this statement.

Response: Reference has been provided.

The general observation is that there is need to consider language usage, thus the authors should not use colloquial language but statements should be supported by references to enhance authenticity of claims. This section can be improved.

Response: Thank you.Language has been edited and whole manuscript has been revised, without altering the technical details mentioned on original manuscript.

Methodology

The methodology for the study is well explained. Perhaps a question one would ask is that since the NDHS used a multi stage stratified sampling, how were cluster and sample design effects dealt with in the analysis of data.

Response: As this manuscript is secondary analysis of NDHS. Methodology summary has only been provided on manuscript. For full methodological details, manuscript has provided reference details. However, as per reviewer suggestion on revised version of manuscript, some additional information on methodology has been incorporated 

Results

Page 12, Line 110: Make a correction to the first sentence ‘Just over half (53.95%) of the participants were of aged 15-29 years, the highlighted word should be ages. Make similar corrections across the result section and ensure that proper language is used for interpretation of results.

Response: Correction has been made as suggested.

Page 14, Line 121: 26.08%, you can’t stand the sentence with a number. Kindly make correction. Moreover its ‘one risk factor’ Not ‘One risk factors’. There is need for language editing to remove typing and grammatical errors.

Response: Thank you. It has been corrected in the manuscript.

There is an over-use and at times misuse of the word ‘similarly’.

Response: Thank you for suggestion. It has been corrected on revised version of manuscript.

For interpretation of table 3, insert confidence intervals for the adjusted prevalence ratios (APR) e.g page 24 line 155, put in confidence intervals together with APRs for primary, secondary or higher education).

Response: Thank you. Confidence interval has been inserted for every significant attributes.

Under the subsection ‘Overweight’ page 24, Line 164-170 there are many typos, please make language corrections. 

Response: It has been corrected 

Discussion 

Page 26 line 201-201 the sentence is not clear….So, this study aim to identify at risk women to possess NCD risk factors…. You need to rephrase the sentence.

Response: It has been rephrased on revised manuscript.

There is no need to categorise the discussion into sub headings. The authors need only to discuss key findings, what are unique and important findings of the study. Only discuss such…what is new and emerging? There are six pages on the discussion section; it shows lack of focus and discussion of salient issues. The discussion section should be concise and provide reasons for any differences in findings when compared to other countries. There should be a subsection on strengths and limitations of the study.

Response: Thank you for suggestions. Whole discussion section has been revisited and edited as per PLoS one format and reviewer suggestions.

References

The authors have not followed the journal’s referencing style. They use the APA style in the reference section, while in the text they use AMA.

Response: It has been corrected.

RESPONSE TO REVIEWER 2

Title: Title is not appropriate as it is, authors should do further analysis or change the title.

Response: Thank you for your very informative suggestion. Manuscript has attempted to provide NCDs related risk factors details on reference to socio-demographic factors. Along with that, manuscript has also attempted to include clustering of NCD risk factors which are available on Demographic and Health survey. So, we believe present title justify the findings presented. However, if reviewer suggests any appropriate alternative topics, it can be considered.

Background: Fair – Can be reworked e.g. lines 62-63 “Thus to tackle with NCDs, the best strategy is to identify and modify the behavioural risk factors that causes NCDs” is not in tandem with the general objective of the work.

Response: It has been rephrased aligning with objective of study.

Methodology: Most poorly documented part of the manuscript. Authors needto follow the systematic reporting of Plos one articles. There are several articles published on NCD risk factors

1. No mention is made of the data collection instrument – was it the stepwise instrument which will allow for comparison with other studies.

Response: Methodology section has been revised. Details has been mentioned. Regarding data collection instrument, since it is Demographic and Health Survey, it has got its own universally standardized data collection tools. So, it is different than that of STEPwise instrument.

2. Variable definitions of outcome variable not precise e.g. lime 90-91 Current tobacco use includes either daily or occasional smoking or use of smokeless tobacco (snuff by mouth, snuff by nose, chewing tobacco and betel quid with tobacco) 

Response: Thank you for suggestion. Definition used here is adopted from table 3.13 of NDHS 2016 .

 Anthropometric measures

Response: Regarding anthropometric measures, BP measurement details have been provided on manuscript. Regarding weight and height, it has been measured by following universally standardized process and procedure. Details are mentioned on DHS Biomarker Field manual that has been mentioned reference list Biological measures

Response: No biological measures have been considered for this manuscript.

Data processing – Very little information

Response: Details of the data processing have been elaborated in the main report that has been cited in the manuscript. We considered writing specific parts of analysis relating to this manuscript to limit the length of manuscript. 

3. Data Analysis – Not clear

Response: Complex survey analysis on stata 15.1 was carried out considering Taylor linearization method of calculation Standard error. Descriptive, bivariate analysis was done to assess relationship. For multivariable analysis multiple poisson regression was considered using APR (adjusted prevalence ratio)

Results

a. Table 1 should be shortened Provinces not likely to be meaningful to an international audience.

Response: Nepal has recently undergone restructuring process moving from unitary to federal structure. Although seven provinces were created by constituent assembly they are yet to be named (only three out of seven are named officially) and referred in official documents with number and we adopted the same in our study. Due to recent restricting process, Nepal lacks information on province wise fashion. So, province wise information may be useful for national as well international policy maker and decision to re-design the program related with NCDs.

b. Reformat Table 2

Response: it has been reformatted.

c. Why was prevalence ratios used and not odds ratios, but were reported as odds ratios “more likely”

Response: Considering interpretation easiness, prevalence ratio is more interpretable and easier to communicate to non-specialists than the odds ratio. Several studies have also recommended the use of prevalence ratio. Reference documents are available here:

• Alternatives for logistic regression in cross-sectional studies: an empirical comparison of models that directly estimate the prevalence ratio.

Barros AJ1, Hirakata VN. (link: https://www.ncbi.nlm.nih.gov/pubmed/14567763)

• The Burden and Determinants of Non Communicable Diseases Risk Factors in Nepal: Findings from a Nationwide STEPS Survey

Krishna Kumar Aryal ,Suresh Mehata ,Sushhama Neupane et.al (link: https://journals.plos.org/plosone/article?id=10.1371/journal.pone.0134834#pone.0134834.ref019)

d. Multivariate of Clustering Risks should be on a separate table – preferably present – mean risk factors. Incidente ratios and robust Standard errors

Response: Thank you for suggestion. Table has been reformatted.

Discussion not acceptable in this current format.

Response: Discussion chapter has been reformatted

---

## [Decision Letter · Decision Letter 1]

25 Nov 2019

PONE-D-19-15867R1

Socio-demographic correlates and clustering of Non-Communicable Diseases risk factors among reproductive aged women of Nepal: Results from Nepal Demographic Health Survey 2016

PLOS ONE

Dear Mr Bista,

Thank you for submitting your manuscript to PLOS ONE. After careful consideration, we feel that it has merit but does not fully meet PLOS ONE’s publication criteria as it currently stands. Therefore, we invite you to submit a revised version of the manuscript that addresses the points raised during the review process.

The reviewers have raised still some concerns on the revised manuscript. Kindly address all those issues raised by the reviewers. 

We would appreciate receiving your revised manuscript by Jan 09 2020 11:59PM. To enhance the reproducibility of your results, we recommend that if applicable you deposit your laboratory protocols in protocols.io, where a protocol can be assigned its own identifier (DOI) such that it can be cited independently in the future. For instructions see: http://journals.plos.org/plosone/s/submission-guidelines#loc-laboratory-protocols

We look forward to receiving your revised manuscript.

Kind regards,

Kannan Navaneetham

Academic Editor

PLOS ONE

Reviewers' comments:

Reviewer's Responses to Questions

**Comments to the Author**

1. If the authors have adequately addressed your comments raised in a previous round of review and you feel that this manuscript is now acceptable for publication, you may indicate that here to bypass the “Comments to the Author” section, enter your conflict of interest statement in the “Confidential to Editor” section, and submit your "Accept" recommendation.

Reviewer #1: All comments have been addressed

Reviewer #2: (No Response)

2. Is the manuscript technically sound, and do the data support the conclusions?

Reviewer #1: Yes

Reviewer #2: Partly

3. Has the statistical analysis been performed appropriately and rigorously? 

Reviewer #1: Yes

Reviewer #2: Yes

4. Have the authors made all data underlying the findings in their manuscript fully available?

Reviewer #1: No

Reviewer #2: Yes

5. Is the manuscript presented in an intelligible fashion and written in standard English?

Reviewer #1: No

Reviewer #2: No

6. Review Comments to the Author

Reviewer #1: Manuscript Review 2

Title: Socio-demographic correlates and clustering of Non-Communicable Diseases risk factors among reproductive aged women of Nepal: Results from Nepal Demographic Health Survey 2016

Manuscript Number: PONE-D-19-15867R1

General Comments

The revised manuscript looks improved in many ways. The authors have addressed many comments which were raised in the previous review. There is general flow of ideas from the introduction to the conclusion section. Consequently this is a much improved version of the manuscript. Most of the sections are improved. However there are minor comments which authors need to address before the manuscript can be accepted for publication. Moreover, there is need for language editing.

Introduction section

Line 61-at the end of the reference and beginning of the sentence which starts with ‘Evidence show…..’ there is need for spacing

Line 64-The entire sentence need to be reconsidered, something is missing, either a conjunctive ‘and’ or a comma before 11.4% of

Line 65-The word ‘indicative’ should be replaced by ‘indication’

Line 70- The word ‘has’ needs to be replaced with ‘have’ since women is plural

Line 72-74-The sentence needs to be rephrased to read better. The word determinates is supposed to be determinants.

Methodology

For me this is section has been well revised and is well presented

Results

Tables look more organized now, and the section is easy to follow. Meanwhile there should be consistency in interpretation of results, for instance you cannot say slightly over one third of women reported multiple NCD risk factors, while only 6.3% reported a single NCD risk factor. These are two different ways of interpreting results in the same sentence and should be avoided. Choose one and stick to it for consistency.

For interpretation of results in table 3 and 4, the Adjusted Prevalence Ratios & Adjusted Risk Ratios are supposed to be in brackets to put emphasis on the interpretation. For instance, in lines 189-191 the sentence is ‘Compared to the women with no education, women who had pursued secondary level of education were (ARR: 0.87; 95% CI:0.77-0.98) times less likely to experience NCD risk factors’

The correct and the conventional way of writing this sentence is;

‘Compared to the women with no education, women who had pursued secondary level of education were less likely (ARR: 0.87; 95% CI:0.77-0.98) to experience NCD risk factors.’

Consider making the correction.

Discussion

This part is well written. However there are some minor comments noted.

Line 208- replace 3 fold with three-fold

Line 235-cf? Make a correction

References

No comments, they follow the journal style.

Reviewer #2: Socio-demographic correlates and clustering of Non-Communicable Diseases risk factors among reproductive aged women of Nepal: Results from Nepal Demographic Health Survey 2016 – Second Review

The manuscript is much improved as the authors have addressed many of the initial concerns. However, the authors still need to address a few more major concerns.

1. The CHOICE of the DATA utilized for their study. Why did the authors use the Nepal Demographic Health Survey 2016 which does not directly address NCD risk factors when Nepal has conducted the stepwise SURVEY which addresses? All the 8 NCD risk factors which enables a more robust exploration of NCD risk factors than the DHS data.

Researchers have investigated prevalence and factors of NCD in Nepal using the steps SURVEY.

i. Aryal KK, Mehata S, Neupane S, Vaidya A, Dhimal M, Dhakal P, et al. (2015) The Burden and Determinants of Non Communicable Diseases Risk Factors in Nepal: Findings from a Nationwide STEPS Survey. PLoS ONE 10(8): e0134834. https://doi.org/10.1371/journal.pone.0134834

ii. Bista B, Mehata S, Aryal KK, Thapa P, Pandey AR, Pandit A, et al. Socio-demographic Predictors 518 of Tobacco Use among Women of Nepal: Evidence from Non Communicable Disease Risk Factors STEPS 519 Survey Nepal 2013. Journal of Nepal Health Research Council. 2015;13(29):14-9. Epub 2015/09/29. 520 PubMed PMID: 26411707.

Even then this work does not lose its merit but the title should change evidence of clustering from this work may be misleading because of missing variables (3 out of 8). Hence whilst clustering may remain in the body of the work as one of the objectives of the study it should be removed from the title

iii. Olawuyi AT, Adeoye IA (2018) The prevalence and associated factors of noncommunicable disease risk factors among civil servants in Ibadan, Nigeria. PLoS ONE 13(9): e0203587.https://doi.org/10.1371/journal. pone.0203587

I suggest the new title should be simply “Socio-demographic correlates of selected Non-Communicable Diseases risk factors among reproductive aged women of Nepal: Results from Nepal Demographic Health Survey 2016”

However, there is need to justify additional evidence above that which has been provided by Aryal and his co-workers. In addition, authors need to emphasise the importance of NCD risk factors among women of reproductive age. Which may be an important contribution of their study.

2. The authors need to employ an English editor – The flow of the written is still poor. For instance, this section in the result could start with the mean age and standard deviation. The description of the table should flow sequentially

Just over half (53.95%) of the participants were d15-29 years. Largest proportions (36.62%) of the participants were from Janjati group (indigenous group). One thirds (33.34%) had no formal schooling while 76.655% of the participants were married. Most of the participants belonged to the Terai belt (49.89%) and rural areas (63.30%). Similarly, 22.43% and 20.92% of participants belonged to richer and the richest wealth quintile. Most of the participants were engaged in agriculture or were self-employed

3. There a need for a ROBUST description of the study area and setting.

The Provinces – their characteristic features, level of development including infrastructure, are rural or urban, level of westernization and epidemiologic/ nutritional transition going on. This will make interpretation and discussion of results more meaningful. This should apply to the ecological region and ethnic group. For example, it is not clear why all the other provinces (Provinces 1 – 5, Karnali etc should have a lower risk for smoking, overweight and hypertension compared to Province 1)

4. DISCUSSION still needs to be worked on

“Our study demonstrated that the proportion of tobacco use was nearly 3 fold higher in 30-40 years age group women which has also been observed in previous studies [23, 24].”

We need to know why tobacco use is higher among older women of reproductive age compared to younger women, what does this imply and what are your recommendations. Not sufficient to state that it has also been observed in previous studies [23, 24].”

5. NEED TO WRITE A STRONG LIMITATION SECTION – in the light of limited variables

6. OTHERS

a. Variable definitions

i. Is it overweight or overweight and obesity

ii. Occupational status . The categories not seem homogenous Agriculture does that mean self employed? What is services?

Generally, authors need to follow the systematic reporting of Plos one articles. There are several articles published on NCD risk factors

Ikeola Adeoye

7. PLOS authors have the option to publish the peer review history of their article (what does this mean?). If published, this will include your full peer review and any attached files.

Reviewer #1: Yes: Dr Mpho Keetile

Reviewer #2: No

---

## [Author Response · Author response to Decision Letter 1]

22 Jan 2020

Reviewer 1:

General Comments

The revised manuscript looks improved in many ways. The authors have addressed many comments which were raised in the previous review. There is general flow of ideas from the introduction to the conclusion section. Consequently this is a much improved version of the manuscript. Most of the sections are improved. However there are minor comments which authors need to address before the manuscript can be accepted for publication. Moreover, there is need for language editing.

Response: English language has been further reviewed.

Introduction section

Line 61-at the end of the reference and beginning of the sentence which starts with ‘Evidence show…..’ there is need for spacing

Response: Thank you for your comments.It has been revised.

Line 64-The entire sentence need to be reconsidered, something is missing, either a conjunctive ‘and’ or a comma before 11.4% of

Response: Thank you for your comments.It has been revised.

Line 65-The word ‘indicative’ should be replaced by ‘indication’

Response: Thank you for your comments.It has been revised.

Line 70- The word ‘has’ needs to be replaced with ‘have’ since women is plural

Response: Thank you for your comments.It has been revised.

Line 72-74-The sentence needs to be rephrased to read better. The word determinates is supposed to be determinants.

Response: Thank you for your comments.It has been revised.

Methodology

For me this is section has been well revised and is well presented

Results

Tables look more organized now, and the section is easy to follow. Meanwhile there should be consistency in interpretation of results, for instance you cannot say slightly over one third of women reported multiple NCD risk factors, while only 6.3% reported a single NCD risk factor. These are two different ways of interpreting results in the same sentence and should be avoided. Choose one and stick to it for consistency.

For interpretation of results in table 3 and 4, the Adjusted Prevalence Ratios & Adjusted Risk Ratios are supposed to be in brackets to put emphasis on the interpretation. For instance, in lines 189-191 the sentence is ‘Compared to the women with no education, women who had pursued secondary level of education were (ARR: 0.87; 95% CI:0.77-0.98) times less likely to experience NCD risk factors’

The correct and the conventional way of writing this sentence is;

‘Compared to the women with no education, women who had pursued secondary level of education were less likely (ARR: 0.87; 95% CI:0.77-0.98) to experience NCD risk factors.’

Consider making the correction.

Response: Thank you for your comments. It has been revised.

Discussion

This part is well written. However there are some minor comments noted.

Line 208- replace 3 fold with three-fold

Line 235-cf? Make a correction

Response: Thank you for your comments.It has been revised.

References

No comments, they follow the journal style.

Overall response: Thank you for comments and suggestions. As per your expert feedback manuscript has been revised. 

Reviewer 2: 

1. The CHOICE of the DATA utilized for their study. Why did the authors use the Nepal Demographic Health Survey 2016 which does not directly address NCD risk factors when Nepal has conducted the stepwise SURVEY which addresses? All the 8 NCD risk factors which enables a more robust exploration of NCD risk factors than the DHS data.

Researchers have investigated prevalence and factors of NCD in Nepal using the steps SURVEY.

i. Aryal KK, Mehata S, Neupane S, Vaidya A, Dhimal M, Dhakal P, et al. (2015) The Burden and Determinants of Non Communicable Diseases Risk Factors in Nepal: Findings from a Nationwide STEPS Survey. PLoS ONE 10(8): e0134834. https://doi.org/10.1371/journal.pone.0134834

ii. Bista B, Mehata S, Aryal KK, Thapa P, Pandey AR, Pandit A, et al. Socio-demographic Predictors 518 of Tobacco Use among Women of Nepal: Evidence from Non Communicable Disease Risk Factors STEPS 519 Survey Nepal 2013. Journal of Nepal Health Research Council. 2015;13(29):14-9. Epub 2015/09/29. 520 PubMed PMID: 26411707.

Even then this work does not lose its merit but the title should change evidence of clustering from this work may be misleading because of missing variables (3 out of 8). Hence whilst clustering may remain in the body of the work as one of the objectives of the study it should be removed from the title.

iii. Olawuyi AT, Adeoye IA (2018) The prevalence and associated factors of noncommunicable disease risk factors among civil servants in Ibadan, Nigeria. PLoS ONE 13(9): e0203587.https://doi.org/10.1371/journal. pone.0203587

Response: Thank you for your suggestion. Regarding clustering study on STEPS survey 2013 data, that has been done by other authors.However, for women only clustering analysis was not done on STEPS survey 2013.Reviewer suggestion is highly appreciable on that regards but sample size(for 15-49 years) for that study is largely small than that of DHS data of 2016.So,we group of authors decided to work on DHS data. Thank you reviewer for your insight,we will definitely plan to work on recent data STEPS survey as a separate paper. In addition, Title has been revised on revised manuscript.

I suggest the new title should be simply “Socio-demographic correlates of selected Non-Communicable Diseases risk factors among reproductive aged women of Nepal: Results from Nepal Demographic Health Survey 2016” 

However, there is need to justify additional evidence above that which has been provided by Aryal and his co-workers. In addition, authors need to emphasise the importance of NCD risk factors among women of reproductive age. Which may be an important contribution of their study. 

2. The authors need to employ an English editor – The flow of the written is still poor. For instance, this section in the result could start with the mean age and standard deviation. The description of the table should flow sequentially

Just over half (53.95%) of the participants were d15-29 years. Largest proportions (36.62%) of the participants were from Janjati group (indigenous group). One thirds (33.34%) had no formal schooling while 76.655% of the participants were married. Most of the participants belonged to the Terai belt (49.89%) and rural areas (63.30%). Similarly, 22.43% and 20.92% of participants belonged to richer and the richest wealth quintile. Most of the participants were engaged in agriculture or were self-employed

3. There a need for a ROBUST description of the study area and setting. 

The Provinces – their characteristic features, level of development including infrastructure, are rural or urban, level of westernization and epidemiologic/ nutritional transition going on. This will make interpretation and discussion of results more meaningful. This should apply to the ecological region and ethnic group. For example, it is not clear why all the other provinces (Provinces 1 – 5, Karnali etc should have a lower risk for smoking, overweight and hypertension compared to Province 1).

Response: Details about study settings is mentioned on full report and reference has been cited in manuscript. So, we have not included extra details on article to reduce bulkiness of manuscript. However, as per reviwer suggestions we have included province related information wherever necessary.

4. DISCUSSION still needs to be worked on 

“Our study demonstrated that the proportion of tobacco use was nearly 3 fold higher in 30-40 years age group women which has also been observed in previous studies [23, 24].” 

We need to know why tobacco use is higher among older women of reproductive age compared to younger women, what does this imply and what are your recommendations. Not sufficient to state that it has also been observed in previous studies [23, 24].” 

Response: Thank you. As per suggestions discussion has been revised wherever necessary 

5. NEED TO WRITE A STRONG LIMITATION SECTION – in the light of limited variables

Response: Thank you. Limitation has been further stated as per suggestions.

6. OTHERS

a. Variable definitions

i. Is it overweight or overweight and obesity 

 Response: Here in manuscript the overweight includes all those participintants whose BMI is greater>24.9 kg/m2.We revised terminology as per reviewer’s suggestions.Thank you.

ii. Occupational status . The categories not seem homogenous Agriculture does that mean self employed? What is services? 

Response: In context of Nepal, majority of agriculture activities are not with motive of business motive.So, this was considered as self-employed and merged with self-employed.

Generally, authors need to follow the systematic reporting of Plos one articles. There are several articles published on NCD risk factors

---

## [Decision Letter · Decision Letter 2]

24 Feb 2020

Prevalence and determinants of non-communicable diseases risk factors among reproductive aged women of Nepal: Results from Nepal Demographic Health Survey 2016

PONE-D-19-15867R2

Dear Dr. Bista,

We are pleased to inform you that your manuscript has been judged scientifically suitable for publication and will be formally accepted for publication once it complies with all outstanding technical requirements.

With kind regards,

Kannan Navaneetham

Academic Editor

PLOS ONE

Additional Editor Comments (optional):

Reviewers' comments:

Reviewer's Responses to Questions

**Comments to the Author**

1. If the authors have adequately addressed your comments raised in a previous round of review and you feel that this manuscript is now acceptable for publication, you may indicate that here to bypass the “Comments to the Author” section, enter your conflict of interest statement in the “Confidential to Editor” section, and submit your "Accept" recommendation.

Reviewer #1: All comments have been addressed

Reviewer #2: All comments have been addressed

2. Is the manuscript technically sound, and do the data support the conclusions?

Reviewer #1: Yes

Reviewer #2: Yes

3. Has the statistical analysis been performed appropriately and rigorously? 

Reviewer #1: Yes

Reviewer #2: Yes

4. Have the authors made all data underlying the findings in their manuscript fully available?

Reviewer #1: No

Reviewer #2: Yes

5. Is the manuscript presented in an intelligible fashion and written in standard English?

Reviewer #1: No

Reviewer #2: Yes

6. Review Comments to the Author

Reviewer #1: The manuscript has been improved and is legible. However, there is need for language editing, and correction of typographical or grammatical errors before publishing the manuscript.

Reviewer #2: I authors have done a lot to address all the issues and the concerns that were raised. Theirs will be one of the evidence provided from the demographic and health surveys which is widely available in most low and middle income countries.

7. PLOS authors have the option to publish the peer review history of their article (what does this mean?). If published, this will include your full peer review and any attached files.

Reviewer #1: Yes: Dr Mpho Keetile

Reviewer #2: No

---

## [Editor Report · Acceptance letter]

3 Mar 2020

PONE-D-19-15867R2 

Prevalence and determinants of non-communicable diseases risk factors among reproductive aged women of Nepal: Results from Nepal Demographic Health Survey 2016 

Dear Dr. Bista:

I am pleased to inform you that your manuscript has been deemed suitable for publication in PLOS ONE. Congratulations! Your manuscript is now with our production department. 

With kind regards,

on behalf of

Professor Kannan Navaneetham 

Academic Editor

PLOS ONE